# Unraveling the association between gut microbiota and chemotherapy efficacy: a two-sample Mendelian randomization study

Zixuan Jia,[1] Xiufeng Liu,[2] Wei Liao[1]

**ABSTRACT**  Emerging evidence has underscored the complex link between gut microbiota and chemotherapy efficacy; however, establishing causality remains elusive due to confounding factors. This study, leveraging bidirectional two-sample Mendelian randomization (MR) analyses, explores the casual relationship between gut microbiota and chemotherapy efficacy. Utilizing genome-wide association study (GWAS) data from the MiBioGen consortium for gut microbiota and IEU Open GWAS for chemotherapy efficacy, we employed genetic variants as instrumental variables (IVs). The inverse variance weighted (IVW) method, weighted median estimator (WME), and MR-Egger regression method were applied, with sensitivity analyses ensuring robustness. Furthermore, we conducted reverse MR analyses between chemotherapy efficacy and identified significant gut microbial taxa. The results indicated that *genus Butyricicoccus* (OR = 3.7908, 95% CI: 1.4464–9.9350, $P = 0.01$), *Dorea* (OR = 3.3295, 95% CI: 1.2794–8.6643, $P = 0.01$), *Hungatella* (OR = 2.6284, 95% CI: 1.0548–6.5498, $P = 0.04$), and *Turicibacter* (OR = 2.5694, 95% CI: 1.0392–6.3526, $P = 0.04$) were positively associated with chemotherapy efficacy using the IVW method. Conversely, *family Porphyromonadaceae* (OR = 0.2283, 95% CI: 0.0699–0.7461, $P = 0.01$) and *genus Eggerthella* (OR = 0.4953, 95% CI: 0.2443–1.0043, $P = 0.05$) exhibited negative associations. WME demonstrated consistent results with IVW method only for *genus Eggerthella* (OR = 0.3343, 95% CI: 0.1298–0.8610, $P = 0.02$). No significant heterogeneity or horizontal pleiotropy was observed. Reverse MR analyses revealed no significant causal effect of chemotherapy on identified gut microbiota. This study sheds light on the intricate relationship between gut microbiota, with a particular emphasis on the *genus Eggerthella*, and chemotherapy efficacy, offering valuable insights for refining cancer treatment strategies.

**IMPORTANCE**  Global advancements in cancer treatment, particularly in chemotherapy, have notably decreased mortality rates in recent years. However, the correlation between gut microbiota and chemotherapy efficacy remains elusive. Our study, emphasizing the role of *genus Eggerthella*, represented a crucial advance in elucidating this intricate interplay. The identified associations offer potential therapeutic targets, contributing to global efforts for enhanced treatment precision and improved patient outcomes. Furthermore, our findings hold promise for personalized therapeutic interventions, shaping improved strategies in the ever-evolving landscape of cancer treatment.

**KEYWORDS**  gut microbiota, chemotherapy efficacy, Mendelian randomization, *genus Eggerthella*

Cancer continues to pose a significant global health challenge; however, mortality rates have steadily decreased over the years due to various factors including lifestyle changes, increased screening, and advancements in therapeutic strategies like surgery,

Address correspondence to Wei Liao, liaowei@sysucc.org.cn.

The authors declare no conflict of interest.

chemotherapy, radiotherapy, and immunotherapy (1, 2). Chemotherapy, in particular, has evolved from single-agent therapy to a multi-agent approach, playing a vital role in cancer treatment (3).

Recent research has underscored the profound influence of gut microbiota on chemotherapy efficacy (4). The gut microbiota, comprising a diverse range of micro-organisms inhabiting the gastrointestinal tract, plays a crucial role in maintaining homeostasis and influencing the development of various diseases in the host (5). Studies have demonstrated that gut microbiota composition can influence immune responses and impact cancer initiation, progression, and response to treatment (6). For instance, Viaud et al. discovered that mice with tumors, particularly those lacking gut micro-biota or subjected to antibiotic treatment targeting Gram-positive bacteria, exhibited resistance to cyclophosphamide (CTX) (7). Yamamura et al. conducted a case-control study, concluding that a high abundance of *Fusobacterium nucleatum* was associated with 5-FU resistance in cancer patients (8). This finding was further supported by *in vivo* and *in vitro* experiments conducted by Zhang et al. (9). Several studies also suggest that the ileal microbiota, such as *Bacteroides fragilis*, may enhance local immune responses and bolster the anti-tumor efficacy of platinum (10, 11). These discoveries underscore promising strategies for leveraging gut microbiota to enhance chemotherapy efficacy and mitigate side effects.

While observational studies have established associations between gut microbiota and chemotherapy efficacy, causality remains elusive. Mendelian randomization (MR) serves as a robust statistical methodology to elucidate causal relationships, utilizing genetic variants associated with the exposure as instrumental variables (IVs) (12, 13). While MR has been applied to explore the links between gut microbiota and various types of cancers (14–16), no relevant MR studies have yet investigated the connection between gut microbiota and chemotherapy efficacy. Given the wealth of data derived from extensive genome-wide association study (GWAS) summary data sets on both gut microbiota and chemotherapy efficacy, MR analysis has emerged as a novel approach to explore the potentially vital causal connections (17).

In our study, we aim to conduct comprehensive two-sample MR analyses in the African American or Afro-Caribbean population and validate the results in multiple ethnic populations to uncover the causal relationship between gut microbiota and chemotherapy efficacy. By leveraging extensive GWAS data sets on both gut microbiota and chemotherapeutic drug responses, we seek to deepen our understanding of this pivotal interplay, potentially informing more effective cancer treatments.

## RESULTS

### Study design

Our study employed a bidirectional two-sample MR approach to investigate the relationship between gut microbiota and chemotherapy efficacy (Fig. 1). Fundamentally, we considered gut microbiota as the exposure and chemotherapy efficacy as the outcome. We meticulously selected SNPs significantly associated with specific gut microbiota taxa as IVs, adhering to stringent inclusion and exclusion criteria. To explore the potential causality, three fundamental assumptions governing the use of IVs must be met: strong correlation with gut microbiota, independence from any confounding variables, and no influence on chemotherapy efficacy through any alternative pathways aside from their effect on gut microbiota (13, 18). MR analyses were conducted across five taxonomic levels (phylum, class, order, family, and genus), accompanied by a series of sensitivity analyses to ensure result reliability. Additionally, reverse MR analyses were performed to address any potential impact of chemotherapy on the causal relationship between identified gut microbiota and chemotherapy efficacy, thus strengthening the robustness of our findings. To assess the generalizability of our results, we validated our findings using chemotherapy-related data across diverse ethnicities, with a particular focus on the identified gut microbiota. Furthermore, we conducted a transcriptomic

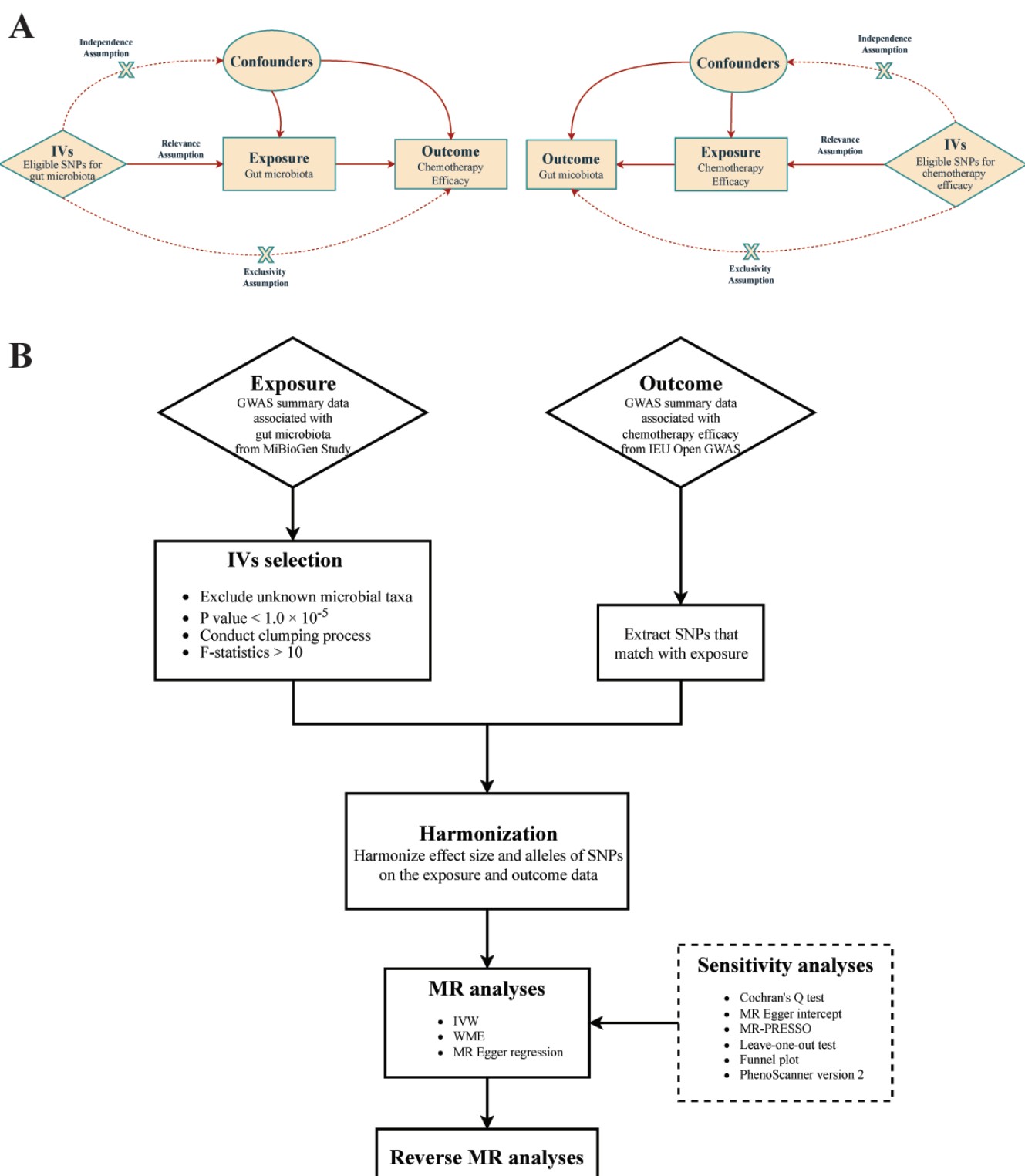

**FIG 1** (A) Schematic diagram showing the present MR analysis and reverse MR analysis. (B) Workflow demonstrating the bidirectional two-sample MR study. Abbreviations: GWAS, genome-wide association study; IVs, instrumental variables; SNPs: single-nucleotide polymorphisms; MR: Mendelian randomization; IVW: inverse-variant weighted; WME: weighted median estimator; MR-PRESSO: MR-pleiotropy residual sum and outlier.

analysis of the identified gut microbiota to delve deeper into its influence on chemotherapy effectiveness.

## Selection of IVs

After a thorough screening process and stringent quality control, we identified 113, 200, 247, 404, and 1,389 single-nucleotide polymorphisms (SNPs) significantly associated with 196 gut microbial traits across five taxonomic levels (phylum, class, order, family, and genus), designated as IVs. All these IVs exhibited significant associations surpassing the stringent locus-wide significance threshold of $P < 1.0 \times 10^{-5}$. For subsequent MR analyses, we extracted essential data, including the effect allele, other allele, beta, standard error (SE), odds ratio (OR), 95% CI (CI), and $P$ value. Notably, all identified IVs demonstrated F-statistics exceeding 10, indicating their robustness and the absence of weak instrument bias.

## Causal effects of gut microbiota on chemotherapy efficacy

In our study, we utilized three approaches: inverse variance weighted (IVW) method, weighted median estimator (WME), and MR-Egger regression methods. Employing the IVW method, we narrowed our selection down to 67 IVs (comprising 11 IVs associated with a single family and 56 IVs associated with six genera), through which we successfully identified seven gut microbial taxa significantly associated with chemotherapy efficacy (Table 1; Table S1). The estimates derived from the IVW analysis revealed that the genetically predicted relative abundance of *genus Butyricicoccus* (OR = 3.791, 95% CI = 1.446–9.935, $P$ = 0.01), *genus Dorea* (OR = 3.330, 95% CI: 1.279–8.664, $P$ = 0.01), *genus Hungatella* (OR = 2.628, 95% CI: 1.055–6.550, $P$ = 0.04), and *genus Turicibacter* (OR = 2.569, 95% CI: 1.039–6.353, $P$ = 0.04) were positively correlated with chemotherapy efficacy. Conversely, IVW estimates indicated that the genetically predicted relative abundance of *family Porphyromonadaceae* (OR = 0.228, 95% CI: 0.070–0.746, $P$ = 0.01), *genus Eggerthella* (OR = 0.495, 95% CI: 0.244–1.004, $P$ = 0.05), and *genus Phascolarctobacterium* (OR = 0.238, 95% CI: 0.074–0.763, $P$ = 0.02) were negatively associated with chemotherapy efficiency (Fig. 2). However, the WME analysis yielded consistent results with the IVW analysis only for *genus Eggerthella* (OR = 0.334, 95% CI: 0.130–0.861, $P$ = 0.02) (Fig. 3). Comprehensive

**TABLE 1** MR analysis results of significant associations between seven gut bacterial taxa and chemotherapy efficacy in the African American or Afro-Caribbean population[a]

| Gut microbiota | Method | nSNPs | beta | SE | OR | 95% CI | P value | Q_P value/ P_intercept |
|---|---|---|---|---|---|---|---|---|
| Family *Porphyromonadaceae* | Inverse variance weighted | 11 | −1.48 | 0.60 | 0.23 | 0.070–0.746 | **0.01** | 0.33 |
| | Weighted median | 11 | −1.26 | 0.81 | 0.28 | 0.057–1.388 | 0.12 | |
| | MR Egger | 11 | 2.27 | 2.22 | 9.69 | 0.124–0.115 | 0.33 | 0.12 |
| Genus *Butyricicoccus* | Inverse variance weighted | 8 | 1.33 | 0.49 | 3.79 | 1.446–9.935 | **0.01** | 0.93 |
| | Weighted median | 8 | 1.14 | 0.63 | 3.13 | 0.906–10.832 | 0.07 | |
| | MR Egger | 8 | 0.70 | 0.89 | 2.02 | 0.353–11.544 | 0.46 | 0.43 |
| Genus *Dorea* | Inverse variance weighted | 11 | 1.20 | 0.49 | 3.33 | 1.279–8.664 | **0.01** | 0.58 |
| | Weighted median | 11 | 0.95 | 0.75 | 2.59 | 0.600–11.167 | 0.20 | |
| | MR Egger | 11 | 1.64 | 1.14 | 5.17 | 0.558–47.894 | 0.18 | 0.68 |
| Genus *Eggerthella* | Inverse variance weighted | 10 | −0.70 | 0.36 | 0.50 | 0.244–1.004 | **0.05** | 0.68 |
| | Weighted median | 10 | −1.10 | 0.48 | 0.33 | 0.130–0.861 | **0.02** | |
| | MR Egger | 10 | 0.35 | 1.57 | 1.42 | 0.065–30.772 | 0.83 | 0.51 |
| Genus *Hungatella* | Inverse variance weighted | 5 | 0.97 | 0.47 | 2.63 | 1.005–6.550 | **0.04** | 0.37 |
| | Weighted median | 5 | 0.71 | 0.65 | 2.03 | 0.570–7.226 | 0.27 | |
| | MR Egger | 5 | 4.77 | 2.87 | 118.40 | 0.426–32930.351 | 0.19 | 0.27 |
| Genus *Phascolarctobacterium* | Inverse variance weighted | 11 | −1.44 | 0.59 | 0.24 | 0.074–0.763 | **0.02** | 0.22 |
| | Weighted median | 11 | −1.18 | 0.82 | 0.31 | 0.062–1.516 | 0.15 | |
| | MR Egger | 11 | −5.59 | 1.79 | 0.00 | 0.000–0.125 | **0.01** | 0.04 |
| Genus *Turicibacter* | Inverse variance weighted | 10 | 0.94 | 0.46 | 2.57 | 1.039–6.353 | **0.04** | 0.78 |
| | Weighted median | 10 | 1.05 | 0.62 | 2.85 | 0.841–9.640 | 0.09 | |
| | MR Egger | 10 | 0.76 | 2.18 | 2.14 | 0.030–153.080 | 0.74 | 0.93 |

[a]nSNPs: number of single-nucleotide polymorphisms; SE, standard error; OR: odds ratio; CI: confidence interval.

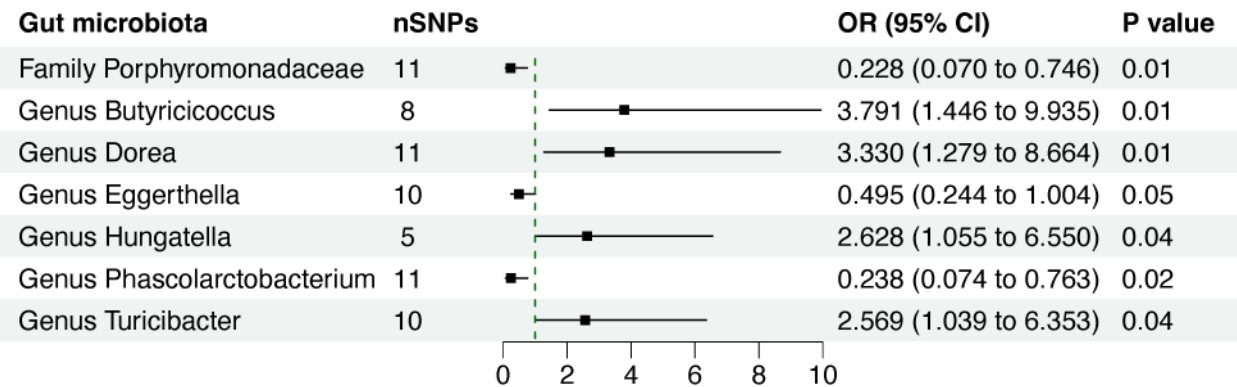

**FIG 2** Forest plot depicting the genetically determined associations between seven gut bacterial taxa and chemotherapy efficacy. Abbreviation: nSNPs: number of single-nucleotide polymorphisms; OR: odds ratio; CI: confidence interval.

results of the MR analyses elucidating the association between 196 gut microbial taxa and chemotherapy efficacy are available in Table S2.

## Sensitivity analyses

Our evaluation of sensitivity adhered to a systematic sequence, aimed at ensuring the credibility and robustness of the MR results. Initially, we performed Cochran's Q test to identify any evidence of heterogeneity among the selected IVs associated with the seven gut microbial taxa, revealing no significant heterogeneity ($P > 0.05$). Subsequently, we conducted pleiotropy analyses, which indicated the absence of horizontal pleiotropic effects among the chosen IVs for six out of the seven gut microbial taxa. Consistent findings from MR-PRESSO showed no significant evidence of outlier SNPs, which could induce horizontal pleiotropy, in six gut microbial taxa, except for *genus Phascolarctobacterium*. Consequently, the specific microbial taxon, *genus Phascolarctobacterium* was excluded from subsequent analyses and discussion. Furthermore, the MR-Egger regression intercepts consistently aligned with the null hypothesis, confirming the absence of horizontal pleiotropy (Table 1). To strengthen the robustness of the identified associations, leave-one-out plots and funnel plots served as additional visual reinforcement (Fig. 4 and 5). In the PhenoScanner database, we identified 1 SNP for gut microbiota nominally associated with chemotherapy efficacy (rs12034718 with lymphocyte count/lymphocyte percentage of white cells/neutrophil percentage of white cells). However, exclusion of this SNP did not alter the pattern of our result.

## Reverse MR analyses

To investigate the possibility of reverse causation, we conducted reverse MR analyses employing SNPs associated with chemotherapy efficacy ($P < 1.0 \times 10^{-5}$) as IVs. These analyses, however, did not reveal any bidirectional causal effect of chemotherapy efficacy on gut microbiota.

## Generalizability assessment across diverse ethnicities

To investigate the universal impact of gut microbiota on chemotherapy efficacy, we utilized GWAS data regarding specific chemotherapeutic drug responses (Ara-C, capecitabine, carboplatin, cisplatin, daunorubicin, etoposide, paclitaxel, and pemetrexed) among diverse ethnic groups (Sub-Saharan African, European, and East Asian). We investigated the presence of comparable causal associations between six identified gut microbial taxa and various chemotherapeutic drug responses (Table S3), revealing a negative correlation between *genus Butyricicoccus* and paclitaxel drug response in the Sub-Saharan African population, suggesting a protective effect (Fig. S1).

To further explore the heterogeneity of causal links between gut microbiota and chemotherapeutic drug responses across different populations, we conducted detailed

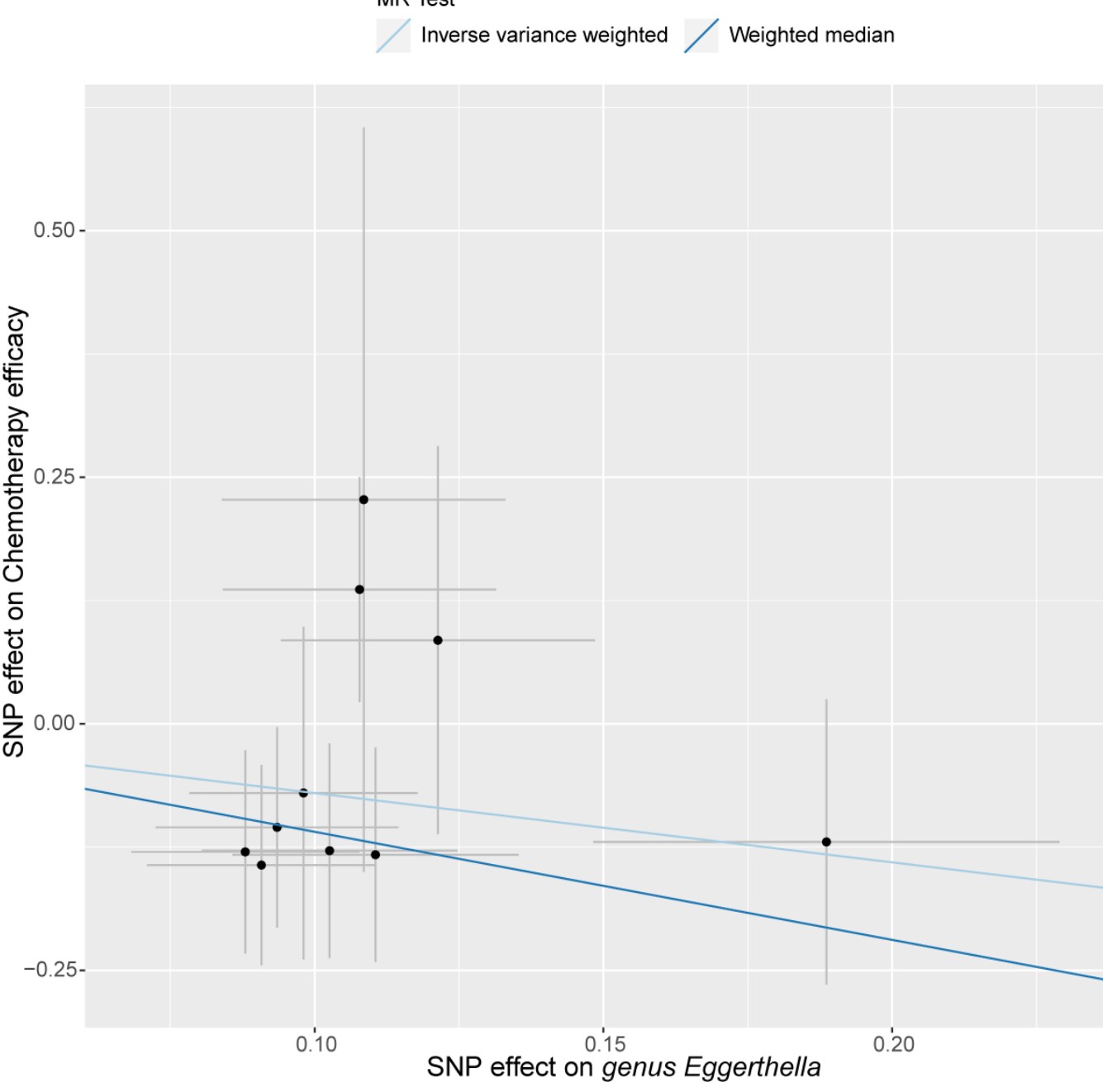

**FIG 3** Scatter plot illustrating the associations of IVs related to *genus Eggerthella* with chemotherapy efficacy. Abbreviations: IVs, instrumental variables; MR, mendelian randomization; SNP, single nucleotide polymorphism.

MR analyses (Fig. S2) incorporating GWAS data of 196 gut microbial taxa and various chemotherapeutic drug responses within the European and East Asian population cohorts. Our results revealed that gut microbial taxa linked to chemotherapeutic drug responses within the European population exhibited no correlation with those within the East Asian population. Within the European population, taxa such as *genus Anaerofilum* and *genus Escherichia* were notably influential, whereas in the East Asian population, taxa such as *order Verrucomicrobiales* and *order Enterobacterales* emerged prominently.

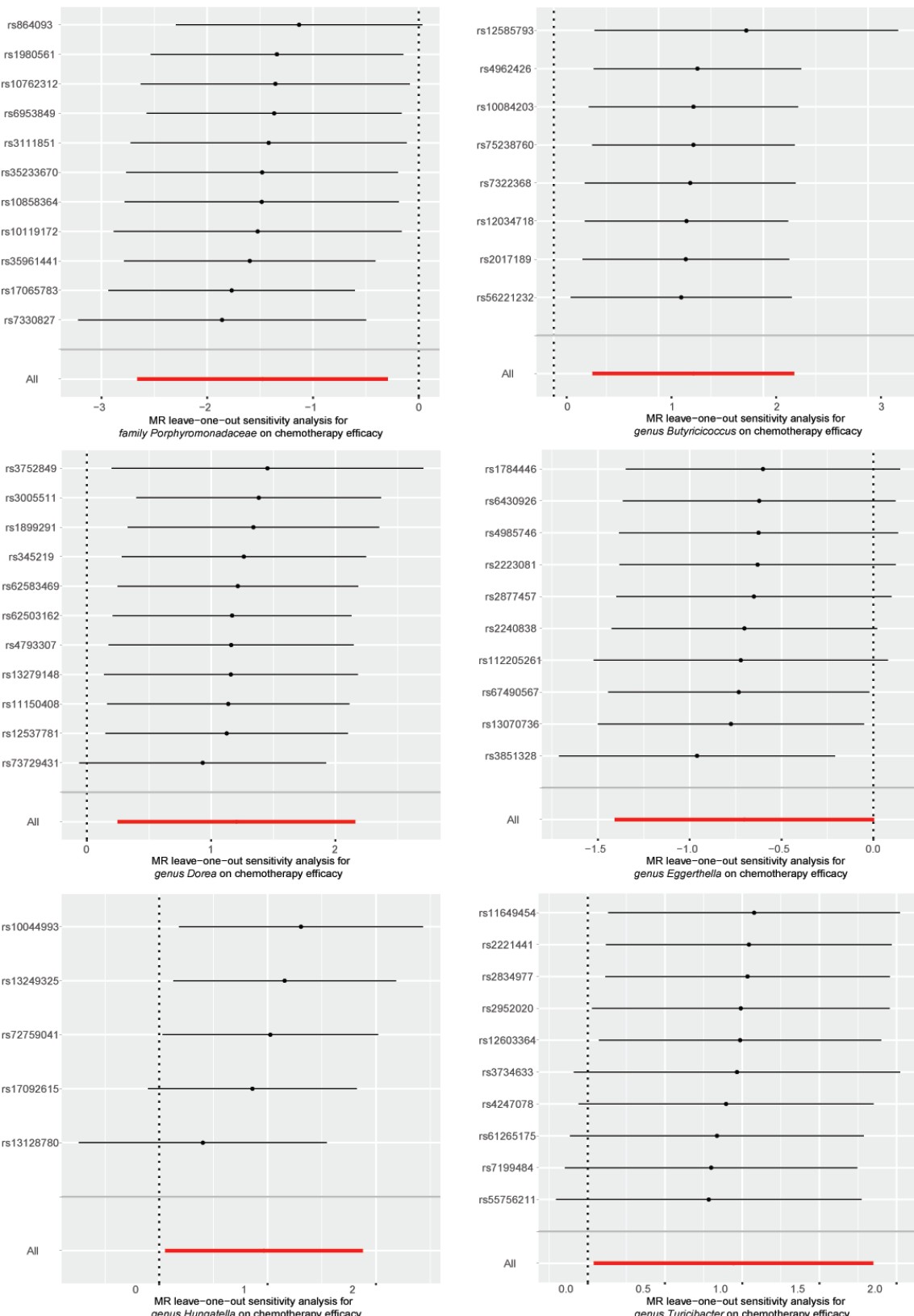

**FIG 4** Leave-one-out sensitivity analyses of single IVs associated with six gut microbial taxa. The red lines are the analysis results of random effects IVW.

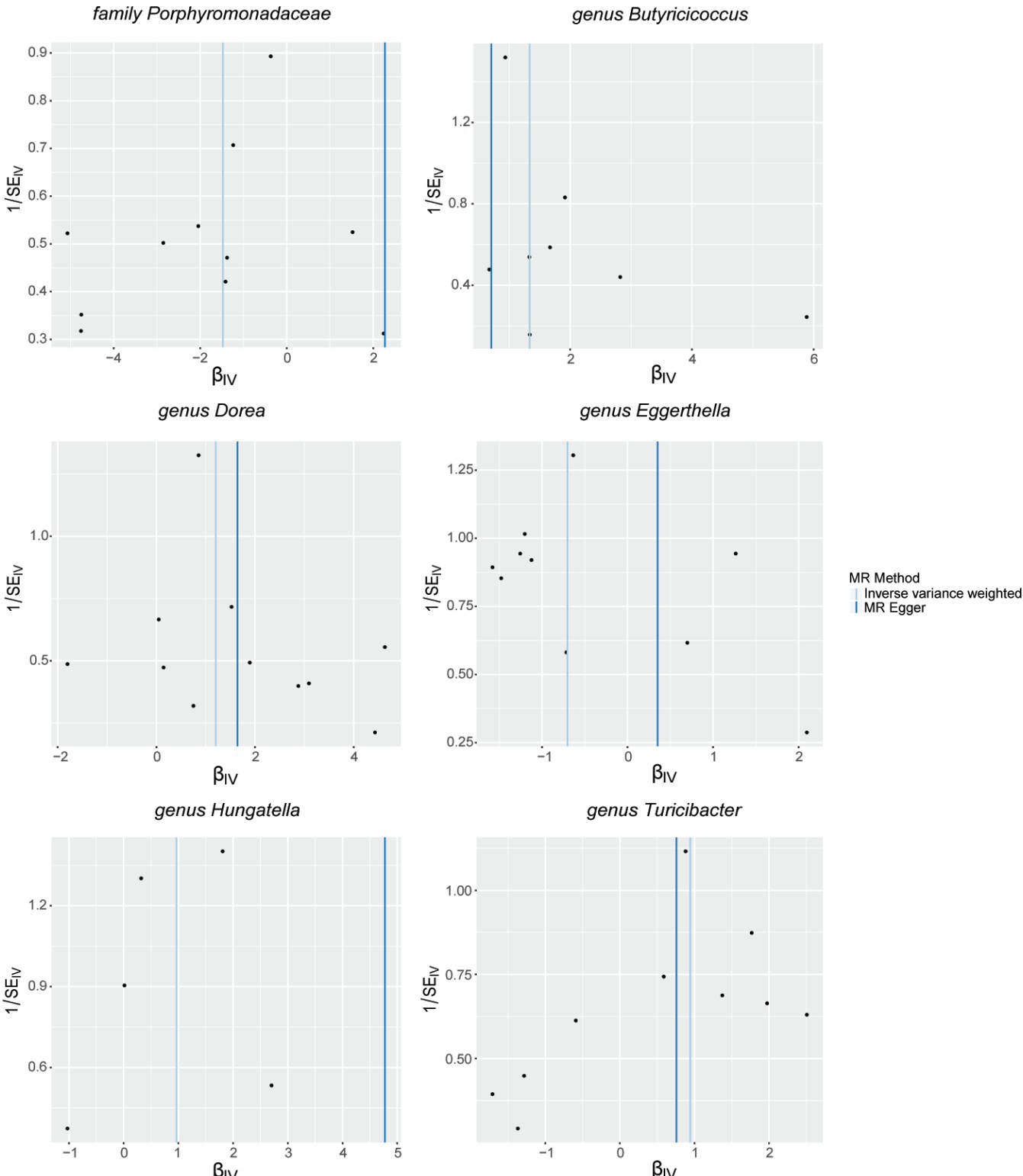

**FIG 5** Funnel plots of single IVs associated with six gut microbial taxa.

## DISCUSSION

Our study conducted a comprehensive exploration of the causal relationship between gut microbiota and chemotherapy efficacy through bidirectional two-sample MR

analyses. Leveraging genetic variants from MiBioGen consortium, the largest GWAS data set of gut microbiota, and data of chemotherapy efficacy from the IEU Open GWAS, we performed MR analyses and discovered genetic associations between six specific gut microbial taxa and chemotherapy efficacy, including *family Porphyromonadaceae* and *genus Butyricicoccus*, *Dorea*, *Eggerthella*, *Hungatella*, and *Turicibacter*, all showing potential for causal relationships with chemotherapy efficacy ($P < 0.05$ using IVW method). We primarily employed IVW, complemented by sensitivity analyses, to ascertain causal relationships between specific gut microbiota and chemotherapy efficacy within the African American or Afro-Caribbean population. A relationship was deemed positive if the IVW $P$ value was less than 0.05, and there was no evidence of pleiotropy or heterogeneity, despite potential insignificance in other MR methodologies. Results with $P$ values less than 0.05, obtained through both IVW and WME, underscored a robust association between gut microbiota and chemotherapy efficacy. Notably, only one gut microbial taxon, *genus Eggerthella*, exhibited such an association.

Utilizing GWAS data on chemotherapy therapeutic drugs across diverse populations, we validated our findings across diverse ethnicities to ensure the robustness and generalizability of our results. However, we found that our results were applicable primarily to the same ethnicity rather than to a broader population. Despite variations in the causal relationships between gut microbiota and chemotherapy efficacy across ethnic groups, our study consistently demonstrated that *genus Butyricicoccus* influenced chemotherapy efficacy in the African population. Additionally, our analysis revealed significant differences in gut microbiota associated with chemotherapy efficacy between the East Asian and European populations. Validation analysis of our findings across diverse populations suggests that our results are primarily applicable to the same African population but may not generalize to other populations. These variations may be attributed to significant differences in the composition of gut microbiota among diverse ethnic groups (19–21), potentially resulting in variations in the effectiveness of the same chemotherapeutic drug across different ethnicities.

Our findings provided novel insights into the association between gut microbiota and chemotherapy efficacy in the African population, especially the negative correlation between *genus Eggerthella* and chemotherapy efficacy, thus highlighting an aspect of gut microbiota involvement in cancer treatment that has been previously overlooked. Although *genus Eggerthella* can be commonly found in the human digestive tract and plays vital ecological roles, research on its relationship with chemotherapy efficacy is limited. Existing investigations have predominantly focused on other microbial genera such as *Lactobacillus* and *Bifidobacterium* (22), with metabolic interactions of *genus Eggerthella* primarily studied in the context of non-chemotherapeutic drugs such as Levodopa and digoxin (23, 24). However, certain child species of *genus Eggerthella* exhibit immunomodulatory capabilities that may influence host immune responses against tumors (25, 26), presenting a promising area for further exploration. Understanding the intricate interplay between gut microbiota, including less-studied taxa like *genus Eggerthella*, and chemotherapy efficacy is essential for optimizing cancer treatment strategies. This knowledge opens avenues for personalized oncology approaches, wherein the incorporation of gut microbial markers into treatment algorithms holds promise. By identifying six specific gut microbial taxa associated with chemotherapy efficacy, our study supports the potential for clinicians to stratify patients based on their likelihood of responding to specific chemotherapy regimens.

The growing interest in gut microbiota, particularly concerning chemotherapy efficacy, highlights its potential role as a modulator of cancer treatment effectiveness. Recent studies have elucidated the influence of microbiota-derived metabolites and pathways on chemotherapy efficacy, thereby illustrating the intricate interplay among gut microbiota composition, immune microenvironment, and clinical outcomes (27–29). In this context, interventions such as dietary modifications, fecal microbiota transplantation (FMT), and probiotic supplementation offer additional avenues to enhance chemotherapy outcomes (30, 31). By reshaping the gut microbial landscape, these

interventions represent innovative strategies to optimize cancer treatment efficacy and improve patient outcomes. Continued research in this field holds the potential to uncover novel therapeutic targets and refine existing treatment strategies, ultimately leading to more effective and personalized approaches to cancer treatment.

The role of gut microbiota in modulating chemotherapy efficacy has attracted increasing attention in recent years. Numerous preclinical and clinical studies have underscored the significant influence of gut microbial composition on the effectiveness of chemotherapy regimens. Studies conducted in murine models have elucidated how certain microbial taxa possess the capability to metabolize chemotherapeutic agents, thereby modulating their bioavailability and therapeutic outcomes (27). Moreover, clinical investigations have consistently demonstrated associations between gut microbial diversity and chemotherapy responses in cancer patients (32). Despite these advancements, the precise mechanisms and causal pathways underlying this association remain incompletely understood. Recent research has shed light on the mechanisms through which gut microbiota influence chemotherapy efficacy. First, gut microbiota can impact chemotherapy efficacy by translocating across the intestinal barrier and subsequently entering secondary lymphoid organs (7, 33). Additionally, the immunomodulatory properties of certain gut microbial taxa have emerged as a focal point since these microorganisms possess the ability to modulate both innate and adaptive immune responses. Some microbes have been found to remodel the tumor microenvironment by eliciting the activation of tumor-specific T cells (7) while some directly interact with inhibitory T-cell receptors (TCR) such as TIGHT and impair natural killer (NK) cell-mediated tumor elimination (34). Furthermore, metabolites generated by specific gut microbial taxa have emerged as key immunomodulatory factors (35). Notably, short-chain fatty acids (SCFAs) such as pentanoate and butyrate, produced via dietary fiber fermentation, have been shown to enhance the proliferation and anti-tumor ability of cytotoxic T lymphocytes (CTLs). SCFAs also have the capacity to upregulate the function of chimeric antigen receptor (CAR) T cells, underscoring their potential role in influencing chemotherapy efficacy (36). Moreover, the enzymatic activity of gut bacterial beta-glucuronidase (GUS) enzymes has been identified as a significant factor impacting chemotherapy efficacy. These enzymes may contribute to the degradation of chemotherapy agents, leading to gastrointestinal toxicity and compromising treatment effectiveness (37, 38). Specifically, diversity and ecological network function of gut microbiota have been implicated in chemotherapy outcomes. For instance, gut microbes associated with inflammation mitigation have been linked to reduced chemotherapy efficacy, whereas taxa associated with colitis have shown correlations with improved response to chemotherapy (11, 27).

Gut microbiota can influence chemotherapy efficacy across various cancer types through various mechanisms. For instance, in rectal cancer, gut microbiota-mediated nucleotide synthesis has been identified as a mechanism that attenuates the response to neoadjuvant chemoradiotherapy (28). Similarly, in HER2-positive breast cancer, gut microbiota has been found to condition the therapeutic efficacy of trastuzumab (39). In line with this, our study delved into transcriptomic analysis to elucidate the pathways through which certain gut microbes impact chemotherapy effectiveness. Through Spearman correlation analysis, we assessed the correlation between the abundance of six microbial taxa in tumor tissues and gene expression levels. The analysis revealed 4231 positively correlated genes (R > 0.3, $P$-value < 0.05) and 261 negatively correlated genes (R < −0.3, $P$-value < 0.05) (Table S4; Fig. S3A). Subsequently, enrichment analysis identified ten pathways primarily enriched with upregulated genes (Fig. S3B). This suggested that the six microbial taxa might influence the clinical efficacy of chemotherapy by modulating immune and protein pathways, including B cell receptor signaling pathway, neutrophil chemotaxis, chemokine–mediated signaling pathway, and positive regulation of small GTPase-mediated signal transduction. Admittedly, empirical studies have found an association between innate-like B cells and the clinical efficacy of gemcitabine plus cisplatin (GP) chemotherapy in nasopharyngeal carcinoma (40).

By uncovering these molecular mechanisms, we aim to conduct subsequent experimental validation studies to further elucidate the role of gut microbiota in chemotherapy response.

Our study presents several distinctive advantages that distinguish it from prior research. Unlike conventional clinical and laboratory trials focused on gut microbiota-chemotherapy associations, our study adopted a pioneering approach by employing MR analyses. This methodological choice significantly mitigates confounding bias and enables a more robust assessment of causality. Moreover, our research ventured into uncharted territory by comprehensively assessing the causal associations between gut microbiota and chemotherapy efficacy. By conducting bidirectional two-sample MR analyses, we extended beyond previous "oncomicrobiota" MR studies, which primarily focused on cancer etiology and risk, to investigate the influence of gut microbiota on chemotherapy efficacy. Furthermore, through the utilization of MR analyses, we cast a wide net in examining gut microbial taxa, ranging from phylum to genus levels. These meticulous taxonomic analyses enhance our theoretical foundations and bolster the framework for future experimental investigations targeting specific bacterial strains. In addition, our study performed comprehensive validation across diverse populations. Notably, the variation in microbes associated with chemotherapy efficacy among different ethnicities suggests that genetic and environmental factors contribute to the composition of the gut microbiome and its subsequent impact on chemotherapy efficacy. This insight underscores the importance of considering population-specific factors in evaluating the role of gut microbiota in chemotherapy efficacy. Finally, our study elucidated the underlying mechanisms of the influence of six specific gut microbial taxa on chemotherapy efficacy through transcriptomic analysis. This mechanistic insight provides a deeper understanding of the interplay between gut microbiota and treatment outcomes, offering potential therapeutic targets for enhancing efficacy.

Nevertheless, we must acknowledge the presence of certain limitations. First, our study predominantly focused on African American or Afro-Caribbean population. To mitigate bias due to population heterogeneity, we validated the main analysis results on gut microbial taxa and overall microbiota in chemotherapy efficacy related GWAS data from various ethnic groups. Future research should aim to include diverse racial and ethnic groups to enhance the robustness and applicability of our findings across populations. Additionally, we relaxed the $P$ value threshold between instruments and exposures to ensure an adequate number of SNPs, which could potentially increase the risk of violating the first assumption of MR design and introduce weak instrumental bias. We mitigated this by conducting SNP clumping to exclude linkage disequilibrium and ensuring that the F statistic for each SNP exceeded 10, indicating the absence of weak instruments in our MR estimation. Furthermore, funnel plot analyses and results from PhenoScanner provided additional robust justification for our IV selection. Nonetheless, despite these precautions, the limited number of available SNPs for some microbial taxa still introduces bias into our results. Furthermore, despite our efforts to minimize confounding factors, including extensive sensitivity analyses, and leveraging PhenoScanner, the presence of unmeasured confounders and residual pleiotropy may still influence our estimates. Finally, our analysis was limited to gut microbial taxa at the phylum to genus levels, potentially overlooking associations at a more specialized taxonomic level, such as the species or strain level. In summary, while our study provides valuable insights into the relationship between gut microbiota and chemotherapy efficacy, these limitations underscore the necessity for a cautious interpretation of our findings and present opportunities for future research to address these challenges and further advance our understanding of this complex interplay.

## MATERIALS AND METHODS

### Data sources

#### Exposure data

To identify SNPs associated with human gut microbiota, we meticulously selected IVs from a GWAS data set provided by the international consortium MiBioGen (https://mibiogen.gcc.rug.nl/, accessed on 27 September 2023). MiBioGen conducted a large-scale multi-ethnic GWAS meta-analysis, expertly integrating 16S ribosomal RNA gene sequencing profiles with genotyping data from a diverse cohort of 18,340 participants across 24 population-based cohorts, with the primary aim of exploring the intricate relationship between autosomal human genetic variants and microbiota composition (41). From this comprehensive GWAS data set, we identified a total of 211 taxa, spanning five distinct taxonomic levels, including nine phyla, 16 classes, 20 orders, 35 families, and 131 genera. After excluding 15 traits associated with unknown families or genera, we utilized 196 bacterial traits for in-depth analyses.

#### Outcome data

GWAS summary-level data of chemotherapy efficacy in the African American or Afro-Caribbean population, as well as data on different chemotherapeutic drug responses in populations including Sub-Saharan African, European, and Asian ethnicities, were obtained from the IEU Open GWAS Project (Table 2).

### Selection of IVs

To ensure the robustness of the data and the reliability of the results, we implemented stringent quality control measures on the SNPs to obtain compliant IVs. The selection of qualified IVs adhered to three fundamental assumptions mentioned above. Initially, SNPs demonstrating a genetic association with each bacterial taxon across various taxonomic levels and meeting the genome-wide significance threshold were identified as potential

**TABLE 2** Summary of outcome data sets utilized in the study

| Data set | Sample size | Ancestry | Links for data download |
|---|---|---|---|
| IEU Open GWAS Project data set | | | |
| Chemotherapy efficacy | 6,194 | African American or Afro-Caribbean | https://gwas.mrcieu.ac.uk/datasets/ukb-e-197_AFR/ |
| Ara-C drug response (AUC) | 90 | East Asian | https://gwas.mrcieu.ac.uk/datasets/ebi-a-GCST90011781/ |
| | 165 | European | https://gwas.mrcieu.ac.uk/datasets/ebi-a-GCST90011785/ |
| | 177 | Sub-Saharan African | https://gwas.mrcieu.ac.uk/datasets/ebi-a-GCST90011793/ |
| Capecitabine drug response (AUC) | 84 | East Asian | https://gwas.mrcieu.ac.uk/datasets/ebi-a-GCST90011782/ |
| | 165 | European | https://gwas.mrcieu.ac.uk/datasets/ebi-a-GCST90011786/ |
| | 175 | Sub-Saharan African | https://gwas.mrcieu.ac.uk/datasets/ebi-a-GCST90011794/ |
| Carboplatin drug response (IC50) | 90 | East Asian | https://gwas.mrcieu.ac.uk/datasets/ebi-a-GCST90011783/ |
| | 168 | European | https://gwas.mrcieu.ac.uk/datasets/ebi-a-GCST90011787/ |
| | 172 | Sub-Saharan African | https://gwas.mrcieu.ac.uk/datasets/ebi-a-GCST90011795/ |
| Cisplatin drug response (IC50) | 90 | East Asian | https://gwas.mrcieu.ac.uk/datasets/ebi-a-GCST90011784/ |
| | 166 | European | https://gwas.mrcieu.ac.uk/datasets/ebi-a-GCST90011788/ |
| | 175 | Sub-Saharan African | https://gwas.mrcieu.ac.uk/datasets/ebi-a-GCST90011796/ |
| Daunorubicin drug response (IC50) | 86 | European | https://gwas.mrcieu.ac.uk/datasets/ebi-a-GCST90011789/ |
| | 173 | Sub-Saharan African | https://gwas.mrcieu.ac.uk/datasets/ebi-a-GCST90011797/ |
| Etoposide drug response (IC50) | 84 | European | https://gwas.mrcieu.ac.uk/datasets/ebi-a-GCST90011790/ |
| | 171 | Sub-Saharan African | https://gwas.mrcieu.ac.uk/datasets/ebi-a-GCST90011798/ |
| Paclitaxel drug response (AUC) | 77 | European | https://gwas.mrcieu.ac.uk/datasets/ebi-a-GCST90011791/ |
| | 87 | Sub-Saharan African | https://gwas.mrcieu.ac.uk/datasets/ebi-a-GCST90011799/ |
| Pemetrexed drug response (AUC) | 84 | European | https://gwas.mrcieu.ac.uk/datasets/ebi-a-GCST90011792/ |
| | 176 | Sub-Saharan African | https://gwas.mrcieu.ac.uk/datasets/ebi-a-GCST90011800/ |

IVs. The threshold was adjusted to the locus-wide significance level ($P < 1.0 \times 10^{-5}$) due to an insufficient number of qualifying SNPs, in accordance with recommendations from relevant research (42–44). Second, to mitigate potential bias arising from strong linkage disequilibrium (LD), we performed a clumping process using a threshold of $r^2 <0.001$ and a 10,000 kb aggregation window, utilizing a reference panel derived from 1000 Genome Project European samples (14). Third, to ensure the representativeness of SNPs, the minor allele frequency (MAF) threshold of SNPs was set at 0.3. Fourth, to ensure that the effect of the SNP on the exposure and the effect of the SNP on the outcome corresponded to the same allele, we conducted harmonization analyses. When palindromic SNPs existed, the forward strand alleles were inferred using allele frequency information. Fifth, to evaluate the strength of the SNPs, F-statistics were calculated. F-statistics gauge the degree of correlation between exposure factors and outcomes, effectively detecting weak IVs. The formula used for F-statistic is F = $R^2 \times$ (N - 2)/(1 R2), where $R^2$ represents the proportion of variability in exposure explained by each instrument, and N is the sample size (45, 46). An F-statistic value exceeding 10 suggests insufficient evidence of bias due to weak IVs (47, 48). Finally, the effect alleles of the exposure and outcome SNPs were harmonized to ensure compatibility. The list of IVs remaining after the removal of pleiotropic ones was used for the subsequent MR analyses. Last, to further strengthen our analysis and mitigate potential sources of bias, we utilized the online analysis software PhenoScanner (version 2; http://phenoscanner.medschl.cam.ac.uk) in the subsequent sensitivity analyses. Sequential removal of SNPs with pleiotropy enabled the assessment of whether their exclusion impacted the structure of the analysis, thereby aiding in the mitigation of potential weak instrument bias and pleiotropy bias.

## MR analyses

Our study employed three main MR methods—the IVW, WME, and MR-Egger regression methods—to assess the causal effects of gut microbiota on chemotherapy efficacy. The IVW method, which assumes that all IVs are valid, synthesizes the effects of all IVs to estimate variant-specific causal effects, thus yielding the most precise estimates (49, 50). Random-effects IVW was applied in cases exhibiting significant heterogeneity ($P < 0.05$), whereas fixed-effects IVW was utilized otherwise (51). The results obtained through IVW would be unbiased in the absence of horizontal pleiotropy influence (52). To mitigate biased IVW results arising from pleiotropy instrumental SNPs, we consequently conducted WME and MR Egger regression methods. These methods offer relatively robust estimates in the presence of pleiotropic effects, albeit with a trade-off of reduced statistical power. WME provides consistent estimates when ≥ 50% of the weight comes from valid IVs (53) and selects the median MR estimate as the causal estimate (54). MR-Egger regression provides accurate estimates under the assumption of instrument strength independent of direct effect (InSIDE), allowing all IVs to be invalid (55). Comprising three parts—testing for directional pleiotropy, identifying a causal effect, and calculating the causal effect—MR Egger regression intercept can assess overall unbalanced horizontal pleiotropy (56).

## Sensitivity analyses

We systematically performed a series of sensitivity analyses to ensure the robustness of the identified causal associations and rule out the effects of confounding bias. Heterogeneity was assessed using Cochran's Q test, identifying IVs as heterogeneous if their associated $P$ value was less than 0.05 (50). We employed a variety of methods, including MR Egger regression intercept, Mendelian Randomization Pleiotropy RESidual Sum and Outlier (MR-PRESSO), leave-one-out analysis, funnel plot analysis, and additional analyses using online analysis software PhenoScanner, to detect potential pleiotropy and mitigate confounding factors. The MR-Egger regression method, grounded in the InSIDE assumption, served as a test for horizontal pleiotropy and evaluated it through its regression intercept (56). We applied the MR-PRESSO outlier test to assess the pleiotropy significance of each IV, whereas the MR-PRESSO global test evaluated the presence

of overall horizontal pleiotropy. SNPs were sequentially removed based on ascending MR-PRESSO outlier test *P*-values, and subsequent MR-PRESSO global test, including weighted regression and computation of the residual sum of squares (RSS), was applied to the remaining SNPs. A significant decrease in RSS ($P < 0.05$) indicated horizontal pleiotropy for the respective SNP (57). Additionally, we conducted related visual analyses. Leave-one-out sensitivity tests on IVs associated with specific gut microbiota involved individually omitting each IV and recalculating MR estimates using the remaining IVs, ensuring that significant findings were not overly influenced by any single IV (58). We also utilized funnel plot analysis not only to visually confirm the absence of weak instrument bias but also to visually identify potential horizontal pleiotropy through asymmetry. Furthermore, we leveraged PhenoScanner to identify IVs associated with confounding factors, aiming to ascertain their minimal impact on our results. Following the exclusion of these IVs, subsequent MR analyses were conducted to assess the consistency with the original results, thereby ensuring the reliability and credibility of our findings.

## Reverse MR analyses

To investigate the potential causal influence of chemotherapy efficacy on the identified significant gut microbiota, reverse MR analyses were conducted. SNPs associated with chemotherapy efficacy ($P < 1.0 \times 10^{-5}$) were employed as IVs.

## Generalizability assessment across diverse ethnicities

To validate the generalizability of our findings across different ethnicities, we included a more ethnically diverse population, encompassing Sub-Saharan African, European, and Asian ethnicities, and conducted MR analyses to assess the influence of identified gut microbiota, as well as the comprehensive set of 196 gut microbial taxa, on various chemotherapeutic drug responses in different ethnic groups. Corresponding sensitivity analyses were also performed to ensure the robustness of the results.

## R software

All statistical analyses were executed in R (version 4.3.1). IVW, WME, and MR-Egger regression methods were conducted using the "TwoSampleMR" package (version 0.5.7). The MR-PRESSO test was implemented using the "MRPRESSO" package (version 1.0).

## Conclusion

In summary, our findings provide genetic evidence of significant relationships between gut microbiota and chemotherapy efficacy in the African populations. Our study underscores the significance of gut microbiota as biomarkers or therapeutic targets for improving chemotherapy efficacy. Understanding this connection may pave the way for the development of more targeted therapeutic interventions for cancer patients.

## ACKNOWLEDGMENTS

We appreciate the participants of all GWAS cohorts included in the present study and the investigators of the IEU Open GWAS project and MiBioGen consortium for sharing related GWAS summary statistics.

This study was financially supported by funding from Natural Science Foundation of Guangdong Province (2022A1515012631).

## AUTHOR AFFILIATIONS

[1]Intensive Care Unit, State Key Laboratory of Oncology in South China, Guangdong Provincial Clinical Research Center for Cancer, Sun Yat-sen University Cancer Center, Guangzhou, China

[2]Biotherapy Center/Melanoma and Sarcoma Medical Oncology, State Key Laboratory of Oncology in South China, Guangdong Provincial Clinical Research Center for Cancer, Sun Yat-sen University Cancer Center, Guangzhou, China

## AUTHOR ORCIDs

Zixuan Jia  http://orcid.org/0009-0008-4903-1744
Wei Liao  http://orcid.org/0000-0002-7945-6726

## DATA AVAILABILITY

The data of gut microbiota and chemotherapy efficacy presented in this study are publicly available. GWAS data set of gut microbiota can be downloaded from international consortium MiBioGen (https://mibiogen.gcc.rug.nl/). GWAS data sets of chemotherapy efficacy can be obtained from the IEU Open GWAS Project using the links listed in Table 2.

## ETHICS APPROVAL

In our study, publicly available GWAS summary statistics were employed. No additional ethics approval or informed consent was deemed necessary for this research.

## ADDITIONAL FILES

The following material is available online.

### Supplemental Material

**Fig. S1 (Spectrum03948-23-s0001.tif).** Scatter plot.
**Fig. S2 (Spectrum03948-23-s0002.tif).** Venn diagrams.
**Fig. S3 (Spectrum03948-23-s0003.tif).** Correlation analysis of six microbial taxa abundance and gene expression levels.
**Supplemental material (Spectrum03948-23-s0004.docx).** Captions for figure and tables.
**Supplemental tables (Spectrum03948-23-s0005.xlsx).** Tables S1 to S4.

### Open Peer Review

**PEER REVIEW HISTORY (review-history.pdf).** An accounting of the reviewer comments and feedback.

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
