## [Reviewer comments · Microbiology Spectrum]

Microbiology Spectrum

Unraveling the Association between Gut Microbiota and Chemotherapy Efficacy: A Two-Sample Mendelian Randomization Study

Zixuan Jia, Xiufeng Liu, and Wei Liao

Corresponding Author(s): Wei Liao, Sun Yat-sen University Cancer Center

Review Timeline:

Submission Date:	November 16, 2023
Editorial Decision:	February 20, 2024
Revision Received:	April 26, 2024
Accepted:	June 20, 2024

Editor: Jonathan Jacobs

Reviewer(s): Disclosure of reviewer identity is with reference to reviewer comments included in decision letter(s). The following individuals involved in review of your submission have agreed to reveal their identity: Ying Yu (Reviewer #2)

Transaction Report:

DOI: <https://doi.org/10.1128/spectrum.03948-23>

Re: Spectrum03948-23 (Unraveling the Association between Gut Microbiota and Chemotherapy Efficacy: A Two-Sample Mendelian Randomization Study)

Dear Dr. Wei Liao:

Thank you for the privilege of reviewing your work. Below you will find my comments, instructions from the Spectrum editorial office, and the reviewer comments.

Thank you for your patience while I searched for two reviewers with the expertise to review this manuscript. This took much longer than I anticipated. Unfortunately, and I am in agreement with the assessment of Review #1, this research needs further work to strengthen the underlying methodology presented. Although the overall approach of using MR for such analysis is novel, as it stands there remain too many unaddressed questions. Please revise the manuscript accordingly and resubmit your work when it's ready.

Revision Guidelines

Sincerely,
Jonathan Jacobs
Editor
Microbiology Spectrum

Reviewer #1 (Public repository details (Required)):

they used a GWAS database.IEU Open GWAS Project

Reviewer #1 (Comments for the Author):

The manuscript presents an innovative approach to understanding the causal relationship between gut microbiota and chemotherapy efficacy using a bidirectional two-sample Mendelian Randomization (MR) analysis. The study leverages genome-wide association study (GWAS) data from the MiBioGen consortium and IEU Open GWAS, employing genetic variants as instrumental variables (IVs). The methodology involves the use of statistical methods like the inverse variance weighted (IVW) method, weighted median estimator (WME), and MR-Egger regression.

1. The choice and strength of IVs in MR analysis are pivotal. Concerns arise regarding the potential weak instrument bias, which could undermine the validity of the findings. The study would benefit from a more robust justification and explanation of the IV selection criteria.
2. A more comprehensive discussion on the statistical methods used, including a rationale for choosing specific MR methods and sensitivity analyses, would enhance the manuscript's clarity and robustness.
3. The study predominantly utilizes data from individuals of European descent, potentially limiting the generalizability of the findings. Incorporating a more ethnically diverse population in the analysis could significantly enhance the applicability of the results across different demographic groups.
4. The manuscript lacks a detailed discussion on the translation of findings into clinical practice, particularly how the identified associations could impact cancer treatment strategies.
5. Further exploration into the biological mechanisms through which the identified microbial taxa influence chemotherapy efficacy is needed for a more comprehensive understanding.
6. The reliance on MR analysis necessitates a thorough discussion of potential confounding factors and alternative pathways. The manuscript would benefit from addressing how these factors were considered and mitigated in the analysis.
7. Additional sensitivity analyses and robustness checks are needed to strengthen confidence in the study's findings. This is particularly important given the potential for weak instrument bias in MR studies.
8. While the study acknowledges some limitations, a more detailed discussion on addressing these in future research would be beneficial. Suggestions for future studies that could validate and extend the findings would add value to the manuscript. The manuscript offers an intriguing approach to examining the relationship between gut microbiota and chemotherapy efficacy. However, to enhance the credibility and impact of the study, addressing the aforementioned methodological concerns, broadening population representation, and providing a more in-depth analysis of the results and their implications are recommended.

Reviewer #2 (Public repository details (Required)):

Line 470-471, The author said that the data presented in this study are publicly available. However, they should specify which public database and if possible, provide the path.

Reviewer #2 (Comments for the Author):

Refer to the attachment.

Dear Editor,

Thank you for forwarding the reviewers' comments on our manuscript entitled "Unraveling the Association between Gut Microbiota and Chemotherapy Efficacy: A Two-Sample Mendelian Randomization Study". We deeply appreciate the thorough evaluation provided by both Reviewer #1 and Reviewer #2 and have conscientiously deliberated upon each aspect raised. Herein, we present a meticulous response to each concern:

Reviewer #1 (Comments for the Author):

The manuscript presents an innovative approach to understanding the causal relationship between gut microbiota and chemotherapy efficacy using a bidirectional two-sample Mendelian Randomization (MR) analysis. The study leverages genome-wide association study (GWAS) data from the MiBioGen consortium and IEU Open GWAS, employing genetic variants as instrumental variables (IVs). The methodology involves the use of statistical methods like the inverse variance weighted (IVW) method, weighted median estimator (WME), and MR-Egger regression.

1. The choice and strength of IVs in MR analysis are pivotal. Concerns arise regarding the potential weak instrument bias, which could undermine the validity of the findings.

The study would benefit from a more robust justification and explanation of the IV selection criteria.

Response: Thanks for your suggestion. Instrumental variables (IVs) refer to single nucleotide polymorphisms (SNPs) significantly associated with exposure variables. In this study, we utilized genetic variants significantly associated with gut microbiota as IVs. The selection of IVs adhered to three criteria (**Line 99-103, Page 6**) (BMJ, 2018) ¹: Firstly, the IVs must demonstrate a strong correlation with the exposure. Secondly, the IVs should remain independent of any confounding variables. Thirdly, the IVs exclusively influence the outcome through their effect on the exposure and not via any alternative pathways. Under these conditions, qualified variants strongly associated with gut microbiota may serve as proxies for estimating potential relationships with chemotherapy efficacy. Consistence with previous credible research and the three basic criteria, we have conducted the following analyses to select IVs and reduce potential weak instrument bias (**Line 375-402, Page 20-21**): (1) Due to the limited number of available SNPs, we selected SNPs with a lenient P value $< 1 \times 10^{-5}$, a threshold widely used in cases with a limited number of SNPs available (Nature genetics, 2019; BMC medicine,2022; JAMA psychiatry, 2019)²⁻⁴; (2) To mitigate potential bias caused by strong linkage disequilibrium (LD) among selected SNPs, we conducted a clumping progress ($r^2 < 0.001$, clumping distance = 10,000 kb), utilizing a reference clumping panel derived from 1000 Genome Project European samples (BMC medicine, 2023)⁵. (3)

To ensure the representativeness of SNPs, the minor allele frequency (MAF) threshold of SNPs was set at 0.3. (4) To ensure that the effect of the SNP on the exposure and the effect of the SNP on the result corresponded to the same allele, we conducted harmonization analyses and the forward strand alleles were inferred using allele frequency information when palindromic SNPs existed. (5) To further assess the strength of IVs, we calculated F-statistics and retained IVs with F-statistics > 10 , indicating the absence of weak instrumental bias (International journal of surgery (London, England), 2024)⁶. Furthermore, we additionally looked up each selected SNP in the PhenoScanner GWAS database (version 2; <http://phenoscanner.medschl.cam.ac.uk>) to assess any previous associations ($P < 1 \times 10^{-5}$) with potential confounding traits (**Line 159-162, Page 9; Figure 1B** in manuscript) (Bioinformatics, 2019; Nature medicine, 2021)^{7,8}. We subsequently assessed the effects of manually removing these SNPs from the MR analyses to rule out possible pleiotropic effects. And removing SNPs did not change the pattern of our initial results, further enhancing the strength and robustness of IVs while mitigating potential weak instrument bias.

2. A more comprehensive discussion on the statistical methods used, including a rationale for choosing specific MR methods and sensitivity analyses, would enhance the manuscript's clarity and robustness.

Response: Thank you for your constructive comments. MR-related studies encompass a

variety MR methods and sensitivity analyses, which differ from study to study. We have meticulously chosen comprehensive methods following thorough comparisons across multiple studies (JAMA psychiatry, 2017; European heart journal, 2022; EBioMedicine, 2023)⁹⁻¹¹, thereby elucidating the rationale behind our methodology. In the METHOD section (**Line 404-420; Page 21-22**), we have provided a more comprehensive and logically structured discussion of the statistical methods employed, including both MR methods and sensitivity analyses, to improve clarity and reliability. To make it more legible, MR methods comprised inverse variance weighted (IVW), weighted median estimator (WME) and MR-Egger regression methods while sensitivity analyses encompassed Cochran's Q test, MR-Egger regression intercept, MR-PRESSO, Leave-one-out and funnel plot analyses, and the utilization of PhenoScanner, all of which were discussed in a more detailed way.

3. The study predominantly utilizes data from individuals of European descent, potentially limiting the generalizability of the findings. Incorporating a more ethnically diverse population in the analysis could significantly enhance the applicability of the results across different demographic groups.

Response: Thanks for the suggestion. In this study, we utilized GWAS summary-level data of chemotherapy efficacy from African American or Afro-Caribbean population, which was provided by the IEU Open GWAS Project. In fact, including data for only one

ethnic group may limit the applicability of the results. To improve the generalizability of our findings, we analyzed the potential relationships between six identified gut microbial taxa and various chemotherapeutic drug responses across different ethnicities (Sub-Saharan African, European, and East Asian), and confirmed the correlation between *genus Butyricoccus* and chemotherapy efficacy in the African population. Unfortunately, we do not observe a clear genotype–phenotype correlation in the European and East Asian populations. Studies have demonstrated significant variations in the composition of gut microbiota among diverse ethnic groups (Nature medicine, 2018; Nature medicine, 2018; Nature reviews. Microbiology, 2021) ^{12–14}. We speculated that ethnic differences in gut microbiome composition may result in findings that may not be applicable across diverse demographic groups. To further explore the correlation between gut microbiota and chemotherapy efficacy in the European and East Asian populations, we utilized bidirectional two-sample MR analyses to investigate the causal relationship. In the European population, gut microbial taxa such as *genus Anaerofilum*, *genus Escherichia*, *genus Streptococcus*, and others played a major role, while in the Asian population, *order Verrucomicrobiales*, *order Enterobacterales*, *genus Lactococcus*, and others were prominent (**Fig 1**). These results were discussed in detail in the manuscript. (**Supplementary Fig S2** in manuscript) (**Line 171-190; Page 9-10**). Overall, through incorporating more ethnically diverse populations in our study, we aimed to improve the generalizability of our results within the same demographic group and offer a more comprehensive demonstration of the applicability of our findings.

Fig 1 Venn diagrams showing the causal relationships between gut microbiota and different chemotherapeutic drug responses in European (A) and East Asian populations (B).

4. The manuscript lacks a detailed discussion on the translation of findings into clinical practice, particularly how the identified associations could impact cancer treatment strategies.

Response: Thanks for your comments. Detailed discussion on translating our findings into clinical practice has been added in DISCUSSION (**Line 223-253; Page 12-13**). We focused on discussing how our research findings can be translated into specific measures in clinical practice, particularly how the identified associations could impact cancer treatment strategies. Our study revealed an association between gut microbiota and chemotherapy efficacy in African population, notably highlighting a negative correlation between *genus Eggerthella* and chemotherapy efficiency. This finding underscores the significance of gut microbiota in cancer treatment, especially in the realm of personalized therapy. Additionally, our study also identified other five specific gut microbial taxa associated with chemotherapy efficacy, providing clinicians with a potential approach to tailor treatment plans based on patients' microbial compositions.

We further discussed the potential roles of gut microbiota in cancer treatment and proposed a range of interventions, including dietary modifications, fecal microbiota transplantation (FMT), and probiotic supplementation, to optimize chemotherapy outcomes. Through these innovative strategies, we can reshape the gut microbiota landscape and improve the prognosis of cancer patients. In summary, our research

provides a crucial theoretical foundation and clinical guidance for the development of cancer treatment strategies, offering potential breakthroughs and advancements in future cancer therapies.

5. Further exploration into the biological mechanisms through which the identified microbial taxa influence chemotherapy efficacy is needed for a more comprehensive understanding.

Response: Thanks for your suggestion. We have conducted transcriptomic analysis to further explore the underlying mechanisms. Recent studies have highlighted the impact of microbial taxa on chemotherapy and immunotherapy efficacy (Nat Rev Cancer, 2022; Nat Rev Clin Oncol, 2023)^{15,16}. Gopalakrishnan V et al. observed that gut microbiota modulates the response to anti-PD-1 immunotherapy in melanoma patients, revealing significant differences in the diversity and composition of gut microbiota between responders and non-responders (Science, 2018)¹⁷. Additionally, *Bacteroides vulgatus*-mediated nucleotide biosynthesis is associated with resistance to preoperative neoadjuvant chemoradiotherapy in locally advanced rectal cancer patients. Nonresponsive tumors are characterized by the upregulation of genes related to DNA repair and nucleoside transport (Cancer Cell, 2023)¹⁸. The evidence suggests that the intestinal microbiota can influence the efficacy of clinical treatments, consistent with our findings. However, the precise mechanism underlying this phenomenon remains unclear.

To assess the biological mechanisms by which identified microbial taxa influence chemotherapy efficacy, we obtained flora abundance and gene expression data from various solid tumors of the digestive system, including colorectal adenocarcinoma, rectal adenocarcinoma, stomach adenocarcinoma, and liver hepatocellular carcinoma (Nature, 2020; TCGA database) ¹⁹. We conducted Spearman correlation analysis to examine the correlation between the abundance of six microbial taxa in tumor tissues and gene expression levels. The analysis revealed 4231 genes positively correlated ($R > 0.3$, $p\text{-value} < 0.05$) and 261 genes negatively correlated ($R < -0.3$, $p\text{-value} < 0.05$) (**Figure 2A**). Subsequently, enrichment analysis identified ten pathways primarily enriched with upregulated genes (**Figure 2B**). This suggested that the six microbial taxa might influence the clinical efficacy of chemotherapy by modulating immune and protein pathways, including B cell receptor signaling pathway, neutrophil chemotaxis, chemokine-mediated signaling pathway, and positive regulation of small GTPase-mediated signal transduction. Admittedly, other empirical studies have found an association between innate-like B cells and the clinical efficacy for gemcitabine plus cisplatin (GP) chemotherapy in nasopharyngeal carcinoma (Nature medicine, 2023) ²⁰. Unfortunately, in this study, we currently lack the conditions to further investigate *in vitro* or *in vivo*. In the future, we intend to elucidate the specific molecular mechanisms underlying this influence for further investigations, both *in vitro* and *in vivo*. This aspect of the content is discussed in detail in DISCUSSION (**Line 254-305; Supplementary Table S4; Supplementary Fig. S3** in manuscript).

Fig 2 Correlation analysis of six microbial taxa abundance and gene expression levels in tumor tissues (A) and gene pathway analysis (B).

6. The reliance on MR analysis necessitates a thorough discussion of potential confounding factors and alternative pathways. The manuscript would benefit from addressing how these factors were considered and mitigated in the analysis.

Response: Thanks for your suggestion. We have included a comprehensive discussion of

potential confounding factors and alternative pathways (**Lines 145-162; Page 8-9**) and ensured the removal of confounding factors to the greatest extent possible. A confounding factor is a variable that is a common cause of both the exposure and the outcome, potentially distorting the relationship between them and leading to erroneous conclusions in MR studies. Mendelian randomization, also known as "Mendelian deconfounding," employs genetic variants as IVs to assess causal relationships between exposures and outcomes. Removing confounding factors is essential to meet the independence assumption in IV selection. Failure to address confounding factors can introduce bias into causal inference, known as confounding bias. We employed a variety of methods to detect confounding factors and alternative pathways, including MR-PRESSO, MR Egger regression intercept, Leave-one-out analysis, funnel plot analysis, and additional analyses using PhenoScanner. Utilizing PhenoScanner allowed us to obtain traits related to IVs (Lymphocyte count; Lymphocyte percentage of white cells; Neutrophil percentage of white cells), aiding in the identification of potential confounding factors or alternative pathways (Bioinformatics, 2019; Nature medicine, 2021)^{7,8}. Following the removal of IVs related to confounding factors, subsequent MR analyses showed no significant changes in outcomes, ensuring the stability and reliability of our results. In conclusion, our study rigorously evaluated potential confounding factors and alternative pathways, emphasizing the importance of controlling for these variables to draw valid conclusions.

7. Additional sensitivity analyses and robustness checks are needed to strengthen confidence in the study's findings. This is particularly important given the potential for weak instrument bias in MR studies.

Response: Thanks for your suggestion. To fortify confidence in the robustness of our study findings, it is imperative to address prevalent sources of bias inherent in MR analyses. Chief among these biases are weak instrument bias and pleiotropy bias. To mitigate these biases comprehensively, we synthesized methods employed in MR-related studies to select IVs and conduct sensitivity analyses, drawing upon established methods and incorporating novel approaches. To mitigate weak instrument bias, we adhered to stringent criteria in the selection of IVs (**Line 374-402; Page 19-21**). To address pleiotropy bias, a suite of sensitivity analyses was employed, encompassing Cochran's Q test, MR Egger intercept, MR-PRESSO, Leave-one-out analysis, Funnel plot analysis, alongside novel analyses facilitated by PhenoScanner (**Line 422-447; Page 22-23**). Of particular note, leveraging PhenoScanner enabled the identification of IVs associated with confounding factors, thus mitigating pleiotropy bias. Subsequently, by excluding these implicated IVs and analyzing the remaining subset, further reduction of weak instrument bias was achieved. Through the amalgamation of previously conducted analyses and the incorporation of novel methodologies, we substantially bolstered confidence in the reliability and validity of our study findings.

8. While the study acknowledges some limitations, a more detailed discussion on addressing these in future research would be beneficial. Suggestions for future studies that could validate and extend the findings would add value to the manuscript.

Response: We appreciate the insightful feedback. In response to the suggestion for a more detailed discussion on addressing these limitations in future research, we have made significant revisions (**Line 329-351; Page 17-18**). Specifically, we have provided additional details regarding the potential biases and constraints encountered in our study, such as the focus on a predominantly African American or Afro-Caribbean population and the implications of a relaxed P value threshold in IVs selection. We have outlined strategies employed to mitigate these limitations, including the validation of main analysis results across diverse ethnic groups and the implementation of stringent criteria for SNP selection and sensitivity analyses to prevent weak instrument bias. Furthermore, we have emphasized the importance of future research endeavors in addressing these challenges to enhance the robustness and generalizability of our findings. Suggestions for future studies include expanding the study population to encompass a broader range of racial and ethnic backgrounds, exploring associations at more specialized taxonomic levels, and employing advanced methodologies to account for potential confounders and pleiotropy. By incorporating these revisions, we believe our manuscript now provides a more comprehensive discussion on addressing limitations and offers valuable insights into avenues for future research to validate and extend our findings, thereby contributing

significantly to the scientific discourse on this topic.

Reviewer #2 (Public repository details (Required)):

1. Line 470-471, The author said that the data presented in this study are publicly available. However, they should specify which public database and if possible, provide the path.

Response: Thanks for the advice. In Data availability statement, we declared that the data of gut microbiota and chemotherapy efficacy presented in this study are publicly available (**Line478-482; Page 24-5**). Detailed description and links of the data can be found in METHOD (**Line 354-372 and Table 2**).

Reviewer #2 (Comments for the Author):

1. The author summarized the results of three different MR, but they should discuss the different outcomes of these three MR.

Response: Thanks for the suggestion. In addressing this query, I would elaborate on two main aspects. Firstly, the discrepancies in results among different MR analysis methods stem from their distinct underlying principles and assumptions. In our manuscript, we provided an in-depth explanation of the principles behind the three MR methods to help

readers comprehend their differences (**Line 404-420; Page 21-22**). Secondly, when analyzing results, we typically don't rely solely on the outcomes of one method. Instead, we integrate results from multiple MR analysis methods and sensitivity analyses to determine a conclusive interpretation. In our study, we considered the significance of results from the IVW method (P value < 0.05) and the presence of heterogeneity or pleiotropy. Even if results from other MR methods may not be significant, if the IVW method indicates positivity and there is no evidence of heterogeneity or pleiotropy, we deem it as a positive outcome (**Line 203-207; 11**). This comprehensive analytical approach aids in a more holistic understanding of study outcomes and facilitates drawing more reliable conclusions.

2. The author's description of the conclusion is quite vague and needs further refinement and clarification.

Response: Thanks for the constructive advice. We have refined and clarified our conclusions as follows (**Line 467-472**): In summary, our findings provide genetic evidence of significant relationships between gut microbiota and chemotherapy efficacy in African populations. Our study underscores the significance of gut microbiota as biomarkers or therapeutic targets for improving chemotherapy efficacy. Understanding this connection may pave the way for the development of more targeted therapeutic interventions for cancer patients.

3. The English needs further improvement.

Response: Thank you for your advice.

In our revised manuscript, we have enhanced language expression and improved spelling, grammar, and clarity to ensure better readability with the help of a native speaker. However, it is essential to acknowledge that due to the optimization of language expression, there may be variations in sentence structures between the revised manuscript and the original version, while preserving the intended meaning. We have chosen not to annotate every variation in the "Marked-Up Manuscript" file to prevent clutter and confusion. Instead, we have highlighted and commented on all alterations excluding language modifications in the "Marked-Up Manuscript" file.

Reference

1. Davies NM, Holmes MV, Davey Smith G. Reading Mendelian randomisation studies: a guide, glossary, and checklist for clinicians. *BMJ*. 2018;362:k601.
doi:10.1136/bmj.k601
2. Sanna S, van Zuydam NR, Mahajan A, et al. Causal relationships among the gut microbiome, short-chain fatty acids and metabolic diseases. *Nat Genet*. 2019;51(4):600-605. doi:10.1038/s41588-019-0350-x
3. Li P, Wang H, Guo L, et al. Association between gut microbiota and preeclampsia-eclampsia: a two-sample Mendelian randomization study. *BMC Med*.

2022;20(1):443. doi:10.1186/s12916-022-02657-x

4. Choi KW, Chen CY, Stein MB, et al. Assessment of Bidirectional Relationships Between Physical Activity and Depression Among Adults. *JAMA Psychiatry*. 2019;76(4):399-408. doi:10.1001/jamapsychiatry.2018.4175
5. Long Y, Tang L, Zhou Y, Zhao S, Zhu H. Causal relationship between gut microbiota and cancers: a two-sample Mendelian randomisation study. *BMC Med*. 2023;21(1):66. doi:10.1186/s12916-023-02761-6
6. Zhang F, Xiong Y, Zhang B. Causal effects of gut microbiota on renal tumor: a Mendelian randomization study. *International Journal of Surgery*. 2024;110(3):1870-1872. doi:10.1097/JS9.0000000000001041
7. Zhou S, Butler-Laporte G, Nakanishi T, et al. A Neanderthal OAS1 isoform protects individuals of European ancestry against COVID-19 susceptibility and severity. *Nat Med*. 2021;27(4):659-667. doi:10.1038/s41591-021-01281-1
8. Kamat MA, Blackshaw JA, Young R, et al. PhenoScanner V2: an expanded tool for searching human genotype–phenotype associations. Kelso J, ed. *Bioinformatics*. 2019;35(22):4851-4853. doi:10.1093/bioinformatics/btz469
9. Zeng R, Jiang R, Huang W, et al. Dissecting shared genetic architecture between obesity and multiple sclerosis. *eBioMedicine*. 2023;93:104647.

doi:10.1016/j.ebiom.2023.104647

10. Hartwig FP, Borges MC, Horta BL, Bowden J, Davey Smith G. Inflammatory Biomarkers and Risk of Schizophrenia. *JAMA Psychiatry*. 2017;74(12):1226-1233.
doi:10.1001/jamapsychiatry.2017.3191
11. Zhou A, Selvanayagam JB, Hyppönen E. Non-linear Mendelian randomization analyses support a role for vitamin D deficiency in cardiovascular disease risk. *Eur Heart J*. 2022;43(18):1731-1739. doi:10.1093/eurheartj/ehab809
12. Deschasaux M, Bouter KE, Prodan A, et al. Depicting the composition of gut microbiota in a population with varied ethnic origins but shared geography. *Nat Med*. 2018;24(10):1526-1531. doi:10.1038/s41591-018-0160-1
13. Gaulke CA, Sharpton TJ. The influence of ethnicity and geography on human gut microbiome composition. *Nat Med*. 2018;24(10):1495-1496.
doi:10.1038/s41591-018-0210-8
14. Fan Y, Pedersen O. Gut microbiota in human metabolic health and disease. *Nat Rev Microbiol*. 2021;19(1):55-71. doi:10.1038/s41579-020-0433-9
15. Wong CC, Yu J. Gut microbiota in colorectal cancer development and therapy. *Nat Rev Clin Oncol*. 2023;20(7):429-452. doi:10.1038/s41571-023-00766-x
16. Fernandes MR, Aggarwal P, Costa RGF, Cole AM, Trinchieri G. Targeting the gut

microbiota for cancer therapy. *Nat Rev Cancer*. 2022;22(12):703-722.

doi:10.1038/s41568-022-00513-x

17. Gopalakrishnan V, Spencer CN, Nezi L, et al. Gut microbiome modulates response to anti-PD-1 immunotherapy in melanoma patients. *Science*. 2018;359(6371):97-103.

doi:10.1126/science.aan4236

18. Teng H, Wang Y, Sui X, et al. Gut microbiota-mediated nucleotide synthesis attenuates the response to neoadjuvant chemoradiotherapy in rectal cancer. *Cancer Cell*. 2023;41(1):124-138.e6. doi:10.1016/j.ccell.2022.11.013

19. Poore GD, Kopylova E, Zhu Q, et al. Microbiome analyses of blood and tissues suggest cancer diagnostic approach. *Nature*. 2020;579(7800):567-574.

doi:10.1038/s41586-020-2095-1

20. Lv J, Wei Y, Yin JH, et al. The tumor immune microenvironment of nasopharyngeal carcinoma after gemcitabine plus cisplatin treatment. *Nat Med*. 2023;29(6):1424-1436.

doi:10.1038/s41591-023-02369-6

Re: Spectrum03948-23R1 (Unraveling the Association between Gut Microbiota and Chemotherapy Efficacy: A Two-Sample Mendelian Randomization Study)

Dear Dr. Wei Liao:

Thank for your patience while your revised manuscript was reviewed and re-assessed. You have done a great job addressing all of the reviewers concerns, as well as my own, and I'm happy to report that your manuscript has been accepted.

I am forwarding it to the ASM production staff for publication immediately. Your paper will first be checked to make sure all elements meet the technical requirements. ASM staff will contact you if anything needs to be revised before copyediting and production can begin. Otherwise, you will be notified when your proofs are ready to be viewed.

Sincerely,
Jonathan Jacobs
Editor
Microbiology Spectrum